# Inactivation Validation of Ebola, Marburg, and Lassa Viruses in AVL and Ethanol-Treated Viral Cultures

**DOI:** 10.3390/v16091354

**Published:** 2024-08-24

**Authors:** Todd Cutts, Anders Leung, Logan Banadyga, Jay Krishnan

**Affiliations:** 1National Microbiology Laboratory, Public Health Agency of Canada, Winnipeg, MB R3E 3R2, Canada; todd.cutts@phac-aspc.gc.ca (T.C.); anders.leung@phac-aspc.gc.ca (A.L.); logan.banadyga@phac-aspc.gc.ca (L.B.); 2Department of Medical Microbiology and Infectious Diseases, University of Manitoba, Winnipeg, MB R3E 0J9, Canada

**Keywords:** AVL, BSL-4, inactivation validation, Ebola virus, Lassa virus, lysis buffer, Marburg virus, RG4 viral families, virus inactivation

## Abstract

High-consequence pathogens such as the Ebola, Marburg, and Lassa viruses are handled in maximum-containment biosafety level 4 (BSL-4) laboratories. Genetic material is often isolated from such viruses and subsequently removed from BSL-4 laboratories for a multitude of downstream analyses using readily accessible technologies and equipment available at lower-biosafety level laboratories. However, it is essential to ensure that these materials are free of viable viruses before removal from BSL-4 laboratories to guarantee sample safety. This study details the in-house procedure used for validating the inactivation of Ebola, Marburg, and Lassa virus cultures after incubation with AVL lysis buffer (Qiagen) and ethanol. This study’s findings show that no viable virus was detectable when high-titer cultures of Ebola, Marburg, and Lassa viruses were incubated with AVL lysis buffer for 10 min, followed by an equal volume of 95% ethanol for 3 min, using a method with a sensitivity of ≤0.8 log_10_ TCID_50_ as the limit of detection.

## 1. Introduction

Viruses are unique among living organisms in that their genetic material can be either RNA or DNA. These nucleic acid molecules are routinely extracted for diagnostic and research purposes from viral cultures, human clinical specimens, and samples from animals and the environment. Although nucleic acid extraction is a routine procedure for many laboratories, for biosafety level 4 (BSL-4) laboratories handling high-consequence pathogens such as Ebola virus (EBOV), Marburg virus (MARV), and Lassa virus (LASV), the verification of viral inactivation in samples to be removed from BSL-4 facilities is crucial to ensure personnel and environmental safety during downstream analyses conducted in lower-biosafety level laboratories.

A variety of methods and commercial kits are available to isolate nucleic acids from samples [1,2,3]. Regardless of what method or kit is used, the essential steps involved in the nucleic acid extraction process remain the same: sample lysis, removal of unwanted biomolecules such as proteins and lipids, and purification/concentration of nucleic acids. These steps are accomplished through a variety of mechanical methods, chemical processes, or a combination of both. When chemical methods are used, lysis buffers containing lytic enzymes, chaotropic compounds, denaturing agents, or detergents are employed to accomplish sample lysis. For instance, the QIAamp^®^ Viral RNA mini kit (Cat. No. 52906, Qiagen Corp., Valencia, CA, USA), which is routinely used in BSL-4 laboratories to isolate viral RNA, employs AVL lysis buffer, which contains 50–70% guanidine thiocyanate (GITC) [4]. GITC lyses cells and viruses, and denatures proteins including nucleases [5,6].

AVL lysis buffer has been used for nucleic acid extraction from a variety of enveloped and non-enveloped viruses contained in samples and cultures. Past studies evaluating the effectiveness of AVL involve viruses such as severe acute respiratory syndrome coronavirus 2 (SARS-CoV-2) [4,7], EBOV [8,9,10], Venezuelan equine encephalitis virus [8,11], Eastern equine encephalitis virus, Western equine encephalitis virus, Rift Valley fever virus, MARV, Dengue virus, West Nile virus [8], Chikungunya virus, La Crosse virus [12], Middle East respiratory syndrome-related coronavirus (MERS) [13], African swine fever virus [14], foot-and-mouth disease virus [15], adenovirus 5, Echovirus 11, and human immunodeficiency virus 1 [16]. A consistent finding across these studies is that AVL lysis buffer treatment alone could not accomplish complete viral inactivation. Additional measures, such as alcohol [9,13], heat [9,17,18] or detergent [6,19] treatment, were required to achieve effective viral inactivation. Therefore, the addition of ethanol to the virus culture–AVL lysis buffer mixture, as per the QIAamp Viral RNA Mini Handbook’s instructions [20], has been implemented before samples can be removed from our BSL-4 laboratory.

The presence of GITC (or other such caustic chemicals) in AVL lysis buffer-treated sample mixtures presents a significant challenge when attempting to detect surviving infectious viral particles using cell cultures. These chemicals, even at low concentrations, cause cytotoxicity [8,12,21] or impair the overall health of the cell cultures, which may render the cells unable to support the growth of any infectious viruses present in the lysed sample. Furthermore, cytotoxicity and chemical-induced alterations in cell morphology cannot be easily distinguished from virus-induced cell lysis or cytopathic effect (CPE), the latter of which is characterized by the rounding of cells, formation of syncytia, or the appearance of inclusion bodies caused by a replicating virus [22].

A variety of approaches have been developed to reduce, neutralize, or remove the effects of toxic chemicals, including neutralization with other suitable chemicals, gel filtration [23], ultracentrifugation to separate out viral particles, prolonged dialysis to remove toxic chemicals [24], and the use of size exclusion centrifugal filter columns. The use of size exclusion spin columns to remove cytotoxic chemicals from virus-containing samples has been successfully employed by several previous studies [4,13,19,21]. Following this approach, our study utilized the Amicon Ultra Centrifugal Filter column to achieve the same goal, removing toxic chemicals from infectious viral particles. Our objective was to evaluate the effectiveness of the standard AVL lysis buffer procedure in inactivating Ebola, Marburg, and Lassa virus cultures, preventing any interference from toxic chemicals by using Amicon spin columns. 

## 2. Materials and Methods

### 2.1. Cell Line, Viruses, and Medium

African green monkey Vero E6 cells (CRL-1586; American Type Culture Collection, Manassas, VA, USA) were used for the initial stock virus culture preparation, and also for viral recovery assays after AVL–ethanol treatment. The cells were maintained in Dulbecco’s modified Eagle’s medium (DMEM, Hyclone SH3024302) supplemented with 10% fetal bovine serum (FBS; Gibco 12484028) and 10 units/mL of penicillin/streptomycin (PS, Gibco 10378016), referred to henceforth as cell culture medium (CCM). Vero E6 cells for virus culture and recovery assays were maintained in DMEM with 2% FBS and 10 units/mL PS, referred to henceforth as virus culture medium (VCM). All Vero E6 cell cultures, whether virus-infected or not, were incubated in a humidified incubator at 37 °C with 5% CO_2_.

### 2.2. Stock Virus Culture Preparation

Ebola virus variant Makona (EBOV/Mak, Ebola virus/H. sapiens-tc/GIN/2014/Makona-C07; GenBank accession no. KJ660348), originally isolated from a clinical sample [25], was genetically modified to express enhanced green fluorescent protein (eGFP) [26]. A stock of EBOV/Mak expressing eGFP (EBOV-eGFP) was prepared by infecting ten T175 flasks of Vero E6 cells at a multiplicity of infection of 0.01. Expression of eGFP, observed under a fluorescent microscope (EVOS^®^ FL digital inverted fluorescence microscope, using EVOS light cube GFP 2.0), was evident in the infected Vero E6 cells at day 3 post-infection, but flasks were harvested 9 days post-infection, once 80% of the cell monolayer exhibited pronounced CPE. Flasks were frozen at −80 °C and thawed the following day, and the contents were clarified by low-speed centrifugation at 5000× *g* for 10 min at 4 °C to pellet cellular debris. A previously published method was used to prepare concentrated stock virus cultures [27]; briefly, the clarified supernatants from all flasks were pooled and overlaid onto a 7 mL 20% sucrose cushion in ultracentrifuge tubes, then spun at 134,000× *g* for 2 h using a SW 32 Ti Rotor (Beckman Coulter) at 4 °C. The resulting viral pellets were suspended in 500 µL of VCM/tube and left overnight at 4 °C. Virus suspensions were pooled, aliquoted, and frozen at −80 °C until needed. The titer of the stock virus, determined by the Reed–Muench method [28], was 8.8 log_10_ TCID_50_/mL.

A Marburg virus isolated from bats, designated as 371Bat [29], was genetically modified to express the Zs Green fluorescent protein (MARV-ZsG) [30] and kindly provided by Jonathan Towner, U.S. Centers for Disease Control and Prevention, Atlanta, GA. A stock culture of concentrated MARV-ZsG (6.95 log_10_ TCID_50_/mL) was prepared in ten T175 Vero E6 cell culture flasks, as detailed above for EBOV-eGFP. Similar to EBOV, green fluorescence was evident in the cell cultures at day 3, but flasks were harvested 14 days post-infection, when 80% of the cell monolayer exhibited pronounced CPE.

Lassa virus Josiah, genetically modified to express GFP (LASV-GFP), was obtained from Jens Kuhn, National Institute of Allergy and Infectious Diseases, National Institutes of Health, Bethesda, MD, USA [31]. A stock culture of concentrated LASV-GFP (9.2 log_10_ TCID_50_/mL) was prepared in ten T175 Vero E6 flasks, as detailed above for EBOV-eGFP. However, unlike EBOV and MARV, LASV failed to yield pronounced CPE in its cell cultures; expression of GFP was evident as early as 1 day post-infection, and flasks were harvested at day 4 post-infection, when 100% of the cell monolayer showed fluorescence.

### 2.3. Removal of Toxic Chemicals Using Amicon Filter Columns

The AVL buffer treatment process typically involves mixing 140 µL of virus culture with 560 µL of AVL lysis buffer, followed by 10 min of incubation. The entire 700 µL mixture is then transferred to a fresh tube containing 560 µL of 95% ethanol and inversion-mixed before removal from the BSL-4 laboratory. To remove toxic chemicals from this mixture, we employed the Amicon Ultra-0.5 mL 100 K Centrifugal Filter column (Cat. No. UFC510008, Millipore Sigma, Massachusetts, USA). These columns, made of regenerated cellulose membrane, are designed to concentrate/purify macromolecules and microorganisms from biological samples while simultaneously allowing the removal of small molecules, such as salts and chemicals [32]. Because the Amicon spin column has a volume limitation of approximately 500 µL, we scaled down the volumes used in our assays proportionately while maintaining the ratios used in the BSL-4 sample removal protocol. A reduced volume of 61 µL of virus culture was mixed with 244 µL of AVL lysis buffer and incubated for 10 min; 244 µL of 95% ethanol was then added to the mixture, which was incubated for another 3 min prior to proceeding with the filter column centrifugation procedure, as described below. 

### 2.4. Control Assays

The following four control assays were performed prior to the virus inactivation assay, to ensure that the residual chemicals from the lysed sample mixture were successfully removed: cytotoxicity control, interference control, low-titer positive control, and negative control. Ten T175 flasks were seeded with Vero E6 cells to obtain 80% confluency on the following day: three flasks each for the cytotoxicity control, low-titer positive control, and interference control, and one flask for the negative control.

Cytotoxicity controls were done to verify the removal of toxic chemicals from the mixture, to ensure no cytotoxicity occurred in the cell cultures. These controls were done in replicates of three; 244 µL of AVL buffer, 61 µL of VCM, and 244 µL of 95% ethanol were mixed and the entire volume of mixture from each replicate was transferred to three Amicon spin columns. The columns were centrifuged at 14,000× *g* until approximately 100 µL of retentate remained in each column. The flow through filtrate from the collection tubes was discarded. An additional 400 µL of VCM was added to each column, mixed by inversion, and centrifuged. The process was repeated three more times for a total of four washes to remove chemical residues. After the final wash, 400 µL of VCM was added to each column retentate, gently mixed by pipetting a few times, and incubated at room temperature for two minutes. Each column was transferred to new collection tubes in an inverted orientation (Figure 1) and centrifuged at 1200× *g* for two minutes to elute into the collection tubes. Eluate from each replicate tube was mixed with 30 mL of VCM, inoculated onto three T175 flasks containing Vero E6 cells, and incubated. The cell monolayer was monitored for abnormalities (cytotoxicity, altered morphology, slow growth) for 14 days in comparison to the untreated cell monolayer (negative control).

Low-titer positive controls were done to demonstrate that an inoculum containing exceptionally low levels of virus can be detected in Vero E6 cell cultures. The stock virus culture was diluted in VCM to ≤10^3^ TCID_50_/mL. In replicates of three, 10 µL of diluted virus culture was mixed with 30 mL of VCM, inoculated onto three T175 flasks, and incubated for 14 days. 

Interference controls were done to determine the effects on viral replication caused by any potential residual chemicals remaining after the Amicon column processing. An ultra-low concentration of each virus, approximately 0.7–0.8 log_10_ TCID_50_ per assay, was added to the interference control. Using a high concentration of virus might fail to reveal subtle changes in viral replication caused by any residual toxic chemicals. A replicate of three eluate–VCM mixtures was prepared, as detailed for the cytotoxicity controls; 10 µL of ≤10^3^ TCID_50_/mL diluted virus culture was added to each mixture and inoculated onto three T175 flasks of Vero E6 cells. If there was no impact by any remaining chemicals on the added virus, then the cell monolayer would be expected to exhibit CPE similar to the low-titer positive controls. 

A negative control was prepared, to have a point of reference for healthy cells that could be compared with the other three controls. CCM was aspirated from a T175 flask, 30 mL of VCM was added, and the flask was incubated for 14 days.

### 2.5. Virus Inactivation Assay and Sub-Passage of EBOV, LASV, and MARV

Six T175 flasks with Vero E6 cells at 80% confluency were prepared as above: three flasks were used for the viral inactivation assay, and one flask each for the cytotoxicity control, low-titer positive control, and negative control. For the viral inactivation assay, in replicates of three, 244 µL of AVL lysis buffer was mixed with 61 µL of EBOV eGFP, LASV eGFP, or MARV zsG stock virus culture and incubated for 10 min, after which 244 µL of 95% ethanol was added, mixed by inverting the tubes, and incubated for another 3 min. The entire volume of inactivation assay mixtures was transferred to corresponding Amicon columns, centrifuged, washed, and eluted, as described above. Eluates from each column were mixed with 30 mL of VCM and inoculated onto the Vero E6 flasks. Cell cultures were monitored for 14 days for signs of viral growth and compared to the control flasks for abnormalities such as cytotoxicity, altered morphology, and slow growth.

Inactivation assay flasks showing no signs of viral growth were sub-passaged for confirmation. Sub-passaging involved transferring 1 mL of medium from each T175 flask to a well containing 3 mL of VCM in a 6-well plate with Vero E6 cells. The inoculated plate was incubated for another 14 days and monitored for fluorescence/CPE. The inactivation assay samples were considered free of infectious virus only if this passage also showed no signs of viral replication. All the flasks and plates that showed CPE/fluorescence were scored as positive (“+”), indicating the presence of virus, or negative (“−”), indicating the absence of detectable virus.

## 3. Results

### 3.1. Control Assays

To assess whether Amicon column filtration could effectively remove the residual chemicals in an AVL buffer-based nucleic acid extraction, a single cytotoxicity control experiment consisting of three replicates was performed. VCM without virus was mixed with AVL lysis buffer and ethanol, processed through Amicon spin columns, and inoculated onto Vero E6 cells. After 14 days, no cytotoxicity was observed on the cell monolayer (Table 1), suggesting that the sample processing through the Amicon columns effectively removed residual chemicals, preventing it from negatively affecting the health of the cell monolayer and generating false positive results.

To assess whether any potential chemical residues remaining after Amicon column processing could interfere with viral replication (interference control), we repeated the above experiment, but spiked the eluate with 1 mL of stock virus diluted to 0.7 log_10_ TCID_50_/mL of MARV-ZsG, 0.8 log_10_ TCID_50_/mL of LASV-GFP, or 0.8 log_10_ TCID_50_/mL of EBOV-eGFP prior to infecting the Vero E6 flasks. After 14 days, CPE and fluorescence were observed (Table 1), indicating that Amicon filtration effectively removed residual chemicals, thereby preventing them from interfering with viral replication, even when inoculated with viral titers equal to their limit of detection. 

A negative control flask, consisting of untreated Vero E6 cells maintained in VCM, exhibited no CPE or fluorescence (Table 1). Low-titer positive control flasks for each virus demonstrated green fluorescence within a few days of incubation after infection (Table 1). 

### 3.2. Inactivation Assay after AVL Lysis Buffer and Ethanol Treatment

This assay was performed after all the control assays had been evaluated and deemed satisfactory, and it was done to determine whether the AVL lysis buffer and ethanol incubation protocol used in our BSL-4 laboratory is sufficient to inactivate high-titer cultures of EBOV, MARV, and LASV. We performed a series of inactivation validation assays. High-titer stock cultures of EBOV-eGFP, MARV-ZsG, or LASV-GFP containing 6.95–9.0 log_10_ TCID_50_/mL of virus were mixed with AVL/ethanol to generate volumes containing 6.47–8.02 log_10_ TCID_50_ of virus per test. Each virus culture was processed through Amicon spin columns to remove toxic chemicals and used to inoculate flasks of Vero E6 cells, as described above. A total of three experiments were conducted for EBOV and LASV, while only two experiments were conducted for MARV. Each experiment consisted of three biological replicates. After 14 days, no CPE or fluorescence was observed in any replicate from any of the initial assays or subsequent passages (Table 2, Table 3 and Table 4), indicating that no detectable virus was present in any of the samples after AVL lysis buffer and ethanol treatment. 

In addition, each replicate experiment also included a simultaneously run low-titer positive control, to demonstrate that low titers of the virus can be detected, and a negative control, to provide a point of reference for cell damage due to any residual toxic chemicals after Amicon column filtration. After 14 days, all low-titer positive control flasks exhibited CPE and fluorescence (Table 2, Table 3 and Table 4). Importantly, they demonstrated the sensitivity of this assay, as we were able to detect as low as ≤0.8 log_10_ TCID_50_ of infectious virus, the limit of detection for our assay. The negative controls did not exhibit any CPE or fluorescence, serving as a visual reference for the other flasks during microscopy and also as a cross contamination indicator (Table 2, Table 3 and Table 4).

## 4. Discussion

Unlike the variety of inanimate media that are routinely used for the recovery and detection of most bacterial species, viruses require living host systems such as embryonated eggs, laboratory animals, or cell cultures for their recovery and detection in samples. However, if such a sample contains any toxic chemicals, these chemicals may negatively affect the health of the embryos [33], animals [34], or cells [13,35]. Toxic chemicals in AVL and other buffers that are required for lysing viral particles to extract RNA molecules present a significant challenge to subsequent assays undertaken to determine the presence of surviving infectious viral particles. These chemicals can cause the death of embryonated eggs, laboratory animals, or cell cultures when they are inoculated with the sample/chemical mixture. When cell cultures are used, the presence of an infectious virus is routinely verified by microscopic observation of virus-induced CPE. Chemicals in the lysate mixture may lyse or damage the cell monolayer, mimicking virus-induced cytolysis or CPE, thus, leading to false positives (indicating that a virus is present when it is not). Alternatively, the same chemicals may damage any remaining virus particles, interfering with the virus’s ability to replicate and producing false negatives.

The use of Amicon centrifugal ultra filtration effectively removed the toxic chemicals from the virus culture/lysis buffer mixture while retaining the viral particles to be subsequently eluted off the filter column, such that we were able to detect virus levels as low as ≤0.8 log_10_ TCID_50_ in our cell culture. In addition, we also diluted the eluates recovered from each column 75-fold in VCM to further reduce the effect of any potential breakthrough toxic chemicals. To confirm that the diluted eluate was sufficiently free of toxic chemicals, we performed cytotoxicity and interference assays. These assays investigated the impact of residual chemicals on the Vero E6 cell monolayer (cytotoxicity assay) and on virus replication (interference assay).

A single experiment was performed consisting of three biological replicates. After 14 days, the cells showed no cytotoxicity or signs of cellular damage, suggesting that the column filtration process, along with the 75-fold dilution, effectively removed the effects of harmful chemicals. Interference assays—which were performed similarly, except with spiked-in low-titer EBOV-eGFP, MARV-ZsG, or LASV-GFP—showed signs of viral replication (CPE) and green fluorescence comparable to that of the low-titer positive controls, indicating that the virus replicated normally at levels equal to the limit of detection of our assay without interference from chemicals. Together, these data suggest that the Amicon spin column procedure is suitable for removing the effects of residual chemicals without affecting the integrity of the Vero E6 cell monolayer, which supports viral replication with a sensitivity equal to that of the low-titer positive control.

To assess whether the long-established AVL lysis buffer and ethanol incubation protocol used in our BSL-4 laboratory is sufficient to inactivate EBOV, MARV, and LASV, we performed a series of inactivation validation assays. High-titer stock cultures of EBOV-eGFP, MARV-ZsG, and LASV-GFP virus were each mixed with AVL/ethanol, generating volumes containing 6.47–8.02 log_10_ TCID_50_ of virus per test. Each virus culture sample was mixed with AVL lysis buffer and ethanol, processed through Amicon spin columns, and used to inoculate Vero E6 flasks. After 14 days, no virus was detected in any of the replicate assays or their subsequent passages, indicating that the AVL lysis buffer and ethanol treatment inactivated EBOV, MARV, and LASV.

One minor limitation of this study is the constraint on sample volume processing. In our standard BSL-4 protocol for AVL extraction, we use 140 µL of virus culture combined with 560 µL of AVL lysis buffer and 560 µL of 95% ethanol. However, the Amicon filter column has a maximum capacity of approximately 500 µL. To adapt to this limitation, we scaled down our volumes to 61 µL of virus culture, 244 µL of AVL lysis buffer, and 244 µL of 95% ethanol. Interestingly, a study by Burton et al. showed that larger 2 mL volumes processed through Vivaspin 2.0 columns, which worked successfully in BSL-2, “did not translate well” in BSL-4 for removing chemicals from the AVL lysis buffer. Subsequently, they also had to resort to smaller 500 µL volumes and Amicon spin columns [19], as we did with our experiments for this study.

The goal of this study was to assess the inactivation efficacy of AVL lysis buffer and ethanol on three high-consequence viruses. Viral inactivation was evaluated by observing the development of green fluorescence and CPE, instead of detecting the viral genome through PCR. Although genome detection is necessary for certain viruses, particularly non-culturable ones, this method cannot distinguish between viable and nonviable viral particles.

While the results indicate that the nucleic acid extraction protocol validated here may also be effective at inactivating other RG4 viruses, all of which are also enveloped viruses, further validation studies are necessary to confirm this effectiveness. Such studies should ideally be conducted with RG4 viruses not only in cultures, but also in clinical samples. Given that there are over two dozen RG4 viruses, spanning nine viral families [36,37]—*Arenaviridae* (e.g., Lassa virus), *Filoviridae* (e.g., EBOV), *Flaviviridae* (e.g., Kyasanur Forest disease virus), *Herpesviridae* (e.g., Cercopithecine herpesvirus 1), *Nairoviridae* (e.g., Crimean–Congo hemorrhagic fever virus), *Orthomyxoviridae* (e.g., Alphainfluenzavirus influenzae H1N1 reconstructed 1918 strain) [38], *Paramyxoviridae* (e.g., Nipah virus), *Poxviridae* (e.g., variola virus), and *Rhabdoviridae* (e.g., Bas-Congo virus) [39]—it is crucial to validate this protocol’s efficacy across this diverse group. This will ensure the accuracy and reliability of the protocol for each specific high-consequence pathogenic virus.

## 5. Conclusions

This study’s findings show that no detectable viable virus was present when high-titer Ebola, Marburg, and Lassa virus cultures were incubated with AVL lysis buffer for 10 min, followed by an equal volume of 95% ethanol for 3 min, using a method with a sensitivity of ≤0.8 log_10_ TCID_50_ as the limit of detection. These results further support the effectiveness of our established protocol in inactivating these high-consequence pathogens, and they support the safe transfer of treated samples from BSL-4 laboratories to lower-biosafety level laboratories for further analysis, thereby facilitating essential ongoing research and diagnostic activities while maintaining stringent safety standards.

The robustness of this inactivation method across three different viruses suggests its potential applicability to other similar high-consequence pathogens, though further testing is required.

## Figures and Tables

**Figure 1 viruses-16-01354-f001:**
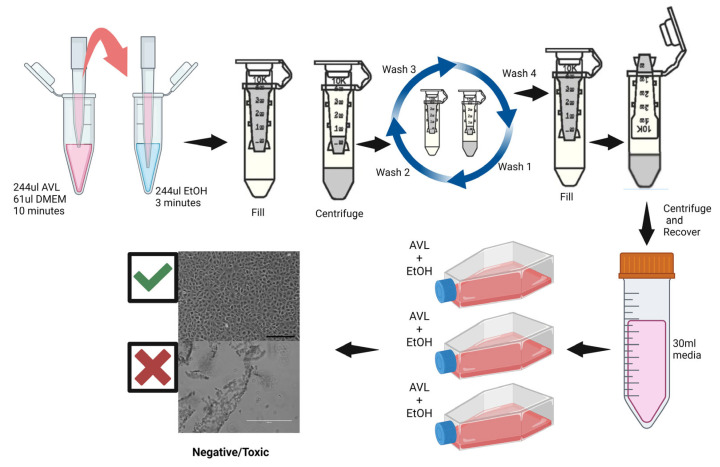
Cytotoxicity control assay. First, 244 µL of AVL lysis buffer, 61 µL of VCM, and 244 µL of 95% ethanol were mixed; the entire volume was transferred to an Amicon Ultra-0.5 mL 100 K spin column; and the column was centrifuged at 14,000× *g* until approximately 100 µL of retentate remained in the column. To wash the column, 400 µL of VCM was added, inverted to mix, and centrifuged, as shown above; this process was repeated three more times. After the final wash, 400 µL of VCM was added to each column, pipetted up and down a few times to mix, and incubated for two minutes. The column was transferred to a new collection tube in an inverted orientation and centrifuged at 1200× *g* for two minutes to elute into the collection tubes. Eluate from each tube was mixed with 30 mL VCM, inoculated onto T175 flasks containing Vero E6 cells, and incubated for 14 days. This figure was created with BioRender.com.

**Table 1 viruses-16-01354-t001:** Removal of the residual toxic chemicals by the Amicon spin column procedure.

	Rep 1	Rep 2	Rep 3
Negative control			
Input virus (log_10_ TCID_50_)	nil	nil	nil
CPE	−	−	−
Fluorescence	−	−	−
Low-titer positive control			
Input virus (log_10_ TCID_50_)	0.7–0.8	0.7–0.8	0.7–0.8
CPE	+	+	+
Fluorescence	+	+	+
Cytotoxicity assay (eluate without virus)			
Input virus (log_10_ TCID_50_)	nil	nil	nil
CPE	−	−	−
Fluorescence	−	−	−
Interference assay (eluate spiked with virus) *			
Input virus (log_10_ TCID_50_)	0.7–0.8	0.7–0.8	0.7–0.8
CPE	+	+	+
Fluorescence	+	+	+

* Eluted samples from Amicon columns were spiked with MARV-ZsG (0.7 log_10_ TCID_50_), LASV-GFP (0.8 log_10_ TCID_50_), or EBOV-eGFP (0.8 log_10_ TCID_50_); + and − indicate presence or absence of CPE or fluorescence after 14 days of incubation

**Table 2 viruses-16-01354-t002:** Inactivation of EBOV-eGFP by AVL lysis buffer and ethanol treatment.

	Experiment 1			Experiment 2			Experiment 3		
AVL/Ethanol	Rep 1	Rep 2	Rep 3	Rep 1	Rep 2	Rep 3	Rep 1	Rep 2	Rep 3
Input virus (log_10_ TCID_50_)	7.37	7.37	7.37	7.37	7.37	7.37	7.57	7.57	7.57
CPE	−	−	−	−	−	−	−	−	−
Fluorescence	−	−	−	−	−	−	−	−	−
Sub-passage (P1)									
CPE	−	−	−	−	−	−	−	−	−
Fluorescence	−	−	−	−	−	−	−	−	−
Low-titer positive control									
Input virus (log_10_ TCID_50_)	0.8	ND	ND	0.8	ND	ND	0.8	ND	ND
CPE	+			+			+		
Fluorescence	+			+			+		
Negative control									
Input virus (log_10_ TCID_50_)	nil	ND	ND	nil	ND	ND	nil	ND	ND
CPE	−			−			−		
Fluorescence	−			−			−		

+ and − indicate presence or absence of CPE or fluorescence after 14 days of incubation; ND: not done.

**Table 3 viruses-16-01354-t003:** Inactivation of MARV-ZsG by AVL lysis buffer and ethanol treatment.

		Experiment 1			Experiment 2	
AVL/Ethanol	Rep 1	Rep 2	Rep 3	Rep 1	Rep 2	Rep 3
Input virus (log_10_ TCID_50_)	6.47	6.47	6.47	6.47	6.47	6.47
CPE	−	−	−	−	−	−
Fluorescence	−	−	−	−	−	−
Sub-passage (P1)						
CPE	−	−	−	−	−	−
Fluorescence	−	−	−	−	−	−
Low-titer positive control						
Input virus (log_10_ TCID_50_)	0.7	ND	ND	0.7	ND	ND
CPE	+			+		
Fluorescence	+			+		
Negative control						
Input virus (log_10_ TCID_50_)	nil	ND	ND	nil	ND	ND
CPE	−			−		
Fluorescence	−			−		

+ and − indicate presence or absence of CPE or fluorescence after 14 days of incubation; ND: not done.

**Table 4 viruses-16-01354-t004:** Inactivation of LASV-GFP by AVL lysis buffer and ethanol treatment.

	Experiment 1			Experiment 2			Experiment 3		
AVL/Ethanol	Rep 1	Rep 2	Rep 3	Rep 1	Rep 2	Rep 3	Rep 1	Rep 2	Rep 3
Input virus (log_10_ TCID_50_)	8.02	8.02	8.02	8.02	8.02	8.02	8.02	8.02	8.02
CPE	−	−	−	−	−	−	−	−	−
Fluorescence	−	−	−	−	−	−	−	−	−
Sub-passage (P1)									
CPE	−	−	−	−	−	−	−	−	−
Fluorescence	−	−	−	−	−	−	−	−	−
Low-titer positive control									
Input virus (log_10_ TCID_50_)	0.8	ND	ND	0.8	ND	ND	0.8	ND	ND
CPE	+			+			+		
Fluorescence	+			+			+		
Negative control									
Input virus (log_10_ TCID_50_)	nil	ND	ND	nil	ND	ND	nil	ND	ND
CPE	−			−			−		
Fluorescence	−			−			−		

+ and − indicate presence or absence of CPE or fluorescence after 14 days of incubation; ND: not done.

## Data Availability

Data are contained within the article.

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
