# Peer review of "Inactivation Validation of Ebola, Marburg, and Lassa Viruses in AVL and Ethanol-Treated Viral Cultures"

_viruses, 2024, doi:10.3390/v16091354_

Round 1
Reviewer 1 Report
Comments and Suggestions for Authors
This study shows that Ebola, Marburg, and Lassa viruses are inactivated by AVL lysis buffer and ethanol treatment. It is essential for researchers working in BSL4 to confirm the inactivation of these viruses using the viral genome isolation kit. However, this confirmation is not novel scientific knowledge, as it is already established that these chemicals inactivate viruses. To enhance the scientific merit of this study, it is recommended to analyze the dose dependence of ethonaol and guanidium thiocyanate (as individual chemicals rather than as mixed solutions in the commercial kit). Different viruses may exhibit varying sensitivities to these chemicals, potentially due to a specific unknown mechanism within the virus, which could be uncovered through this research.
Author Response
This study shows that Ebola, Marburg, and Lassa viruses are inactivated by AVL lysis buffer and ethanol treatment. It is essential for researchers working in BSL4 to confirm the inactivation of these viruses using the viral genome isolation kit. However, this confirmation is not novel scientific knowledge, as it is already established that these chemicals inactivate viruses. To enhance the scientific merit of this study, it is recommended to analyze the dose dependence of ethanol and guanidium thiocyanate (as individual chemicals rather than as mixed solutions in the commercial kit). Different viruses may exhibit varying sensitivities to these chemicals, potentially due to a specific unknown mechanism within the virus, which could be uncovered through this research.
Response: Thank you for your valuable feedback. While we appreciate the suggestion to analyze the dose-dependence of ethanol and guanidium thiocyanate individually, the primary objective of our study was to validate the BSL-4 protocol to assess the efficacy of commercially available AVL lysis buffer and ethanol in inactivating high consequence viruses. Moreover, commercial formulations often include proprietary components and concentrations that contribute to viral inactivation, which may also include unlisted active ingredients. Identifying and testing individual components could limit the applicability of our findings and potentially overlook synergistic effects. Our results demonstrate that the AVL lysis buffer and ethanol treatment effectively inactivates Ebola, Marburg, and Lassa viruses. We believe this information is essential for researchers working in BSL4 settings. Further research into the specific mechanisms of viral inactivation by various known components in commercially available preparations could be a valuable avenue for future studies.
Reviewer 2 Report
Comments and Suggestions for Authors
This paper by Cutts et al investigate a very important issue in virology and especially research regarding BSL-4 pathogens, complete inactivation of viruses. Several methods for inactivation of viruses are available and some of them include using chemicals which are and can be toxic to cells for example. In the current paper the authors have investigated inactivation of Ebola, Marburg and Lassa viruses by using AVL-buffer and ethanol and at the same time examine if the use of the Amicon Ultra Centrifugation Filter column would eliminate the cytotoxic chemicals from the infectious viruses.
The results verify what have been shown before, AVL-buffer and ethanol inactivates viruses and by using the filter the chemicals are removed and causes no cytotoxicity to the cell culture. This confirms AVL-buffer + ethanol as a safe inactivation method for viruses.
Minor comments:
Line 67-70: Here you list several methods that have been used to remove toxic chemicals on cell cultures but it is unclear if this methods have been successful and the references are missing.
Line 106-107: Here you write that the titer of the titer of the EBOV-GPF was 8.8 log10 TCID50/ml. Please specify what method you used to determine the titer (and I assume the other virus titers).
Line 250: How was the limit of detection (≤0.8 log10 TCID50) determined?
Please look over the tables and rearrange the plus and minus signs so that they are in a straight column. For example, the minus sign in table 2, experiment 1, rep 2 and 3 are not directly under their rep.
In addition, the tables are not showing the results from the subsequent passages. Since this is a part of the results and the test of inactivation, please add.
One control that would have been interesting would have verified the presence of virus (active or inactive) in the inoculum is the detection of viral RNA. Why wasn’t PCRs conducted on the inactivated samples to verify presence of virus? Pleas add to discussion.
Author Response
This paper by Cutts et al investigate a very important issue in virology and especially research regarding BSL-4 pathogens, complete inactivation of viruses. Several methods for inactivation of viruses are available and some of them include using chemicals which are and can be toxic to cells for example. In the current paper the authors have investigated inactivation of Ebola, Marburg and Lassa viruses by using AVL-buffer and ethanol and at the same time examine if the use of the Amicon Ultra Centrifugation Filter column would eliminate the cytotoxic chemicals from the infectious viruses.
The results verify what have been shown before, AVL-buffer and ethanol inactivates viruses and by using the filter the chemicals are removed and causes no cytotoxicity to the cell culture. This confirms AVL-buffer + ethanol as a safe inactivation method for viruses.
Minor comments:
Line 67-70: Here you list several methods that have been used to remove toxic chemicals on cell cultures but it is unclear if this methods have been successful and the references are missing.
References added
Line 106-107: Here you write that the titer of the titer of the EBOV-GPF was 8.8 log10 TCID50/ml. Please specify what method you used to determine the titer (and I assume the other virus titers).
Titer in TCID50 was determined by the Reed-Muench method, we have inserted a line along with the method’s reference.
Line 250: How was the limit of detection (≤0.8 log10 TCID50) determined?
Stock virus preparations with known titers were further diluted to obtain approximately 3 log10 TCID50/ml. From each preparation, 10 µl was added to the assay, resulting in an ultra-low titer of about 1 log10 TCID50 per assay. By comparing the actual titer of this ultra-low dilution to the expected value, we determined the minimum detection limit of the virus needed to produce a positive result (CPE/fluorescence) in the low-positive control assay. We then used the same amount of virus, at its minimum detection limit, in the interference control assay to identify any interference from residual chemicals (Table 1). Using ultra-low titers of the virus allowed us to detect even minimal interference from toxic chemicals, whereas higher titers of virus in the interference control assays could obscure low-level interference from residual chemicals. Additional text added to the manuscript to make this point clear
Please look over the tables and rearrange the plus and minus signs so that they are in a straight column. For example, the minus sign in table 2, experiment 1, rep 2 and 3 are not directly under their rep.
Done. The manuscript template provided by the journal is extremely useful; however, predicting the precise alignment of the symbols in the final PDF remains challenging.
In addition, the tables are not showing the results from the subsequent passages. Since this is a part of the results and the test of inactivation, please add.
Done
One control that would have been interesting would have verified the presence of virus (active or inactive) in the inoculum is the detection of viral RNA. Why wasn’t PCRs conducted on the inactivated samples to verify presence of virus? Pleas add to discussion.
While PCR can detect viral nucleic acids, it does not differentiate between infectious and non-infectious virus. Based on our previous experience, there is often minimal variation in PCR signal between positive controls and samples treated with AVL lysis buffer and ethanol. As such, we opted not to include PCR in this study, as it would not provide conclusive evidence of viral inactivation. Additional text added to the discussion section.
Reviewer 3 Report
Comments and Suggestions for Authors
The manuscript by Cutts et al outlines the validation of AVL/ETOH inactivation of EBOV, MARV, and LASV. The manuscript is well written and organized. Most of the text and experiments are focused on removal of the toxic chemicals contained in in AVL buffer by Amicon Ultra Centrifugal Filter columns. Ultimately the manuscript outlines how to validate inactivation processes for viral cultures. Researchers in high containment will find this information very useful.
Two items should be addressed:
1. Lines 165-171. This paragraph should be rewritten so that it is clear that the of virus was spiked into the elutate. I did not fully understand the process until reading table 1.
2. The inactivation assay for EBOV, MARV and LASV should be described in the materials and methods section.
Author Response
The manuscript by Cutts et al outlines the validation of AVL/ETOH inactivation of EBOV, MARV, and LASV. The manuscript is well written and organized. Most of the text and experiments are focused on removal of the toxic chemicals contained in in AVL buffer by Amicon Ultra Centrifugal Filter columns. Ultimately the manuscript outlines how to validate inactivation processes for viral cultures. Researchers in high containment will find this information very useful.
Two items should be addressed:
- Lines 165-171. This paragraph should be rewritten so that it is clear that the of virus was spiked into the eluate. I did not fully understand the process until reading table 1.
Done, we thank you for this feedback.
- The inactivation assay for EBOV, MARV and LASV should be described in the materials and methods section.
We believe that Section 2.5 adequately outlines our methodology, referencing previous sections to avoid redundancy. To enhance clarity, we have included names of the viruses tested at the beginning of Section 2.5.
Round 2
Reviewer 1 Report
Comments and Suggestions for Authors
This result is informative only to scientists working in BSL4 laboratories. The number of such scientists is limited, due to the scarcity of BSL4 facilities worldwide. The authors suggested that commercial buffers may contain unlisted chemicals. If this is the case, this study demonstrates the impact of unlisted chemicals on the viruses. It is not scientific.